# Air Pollutant Traceability Based on Federated Learning of Edge Intelligent Perception Agents

**DOI:** 10.3390/s25196119

**Published:** 2025-10-03

**Authors:** Jinping Xue, Xin Hu, Qiang Liu, Congbo Yin, Peitao Ni, Xinyu Bo

**Affiliations:** 1School of Mechanical Engineering, University of Shanghai for Science and Technology, Shanghai 200093, China; cicixjp0626@outlook.com (J.X.); 2454324@tongji.edu.cn (X.H.); liuq1984520@163.com (Q.L.); boxinyu@ctticsh.cn (X.B.); 2Jiandong Vocational and Technical College, Changzhou 213000, China; scaler_1988@163.com

**Keywords:** Federated Learning, long short-term memory, edge intelligent perception agent, environmental perception, air pollutant traceability

## Abstract

Tracing the source of air pollution presents a significant challenge, especially in densely populated urban areas, because of the unpredictable and complex nature of aerodynamics. To address this issue, intelligent lamp posts have been developed with smart sensors and edge computing capabilities. These lamp posts serve as nodes in the EIPA (Edge Intelligent Perception Agent) network within urban campuses. These lamp posts aim to track air pollutants by employing a tracking algorithm that utilizes big data learning and Gaussian diffusion models. This approach focuses on monitoring the quality of urban air and identifying pollution sources, rather than relying solely on traditional CFD simulations for air pollution dispersion. The algorithm comprises three primary components: (1) the Federated Learning framework built on the EIPA system; (2) the LSTM model implemented on the edge nodes of the EIPA system; and (3) a genetic algorithm utilized for optimizing the model parameters. By using CFD simulations in a simulated city park, training data on air dynamic movements is gathered. The usefulness of the method for tracing air pollutants based on federated learning of edge intelligent perception agents is demonstrated by the outcomes of algorithm training. Experimental results show that, compared to the traditional genetic algorithm (GA) and LSTM + genetic algorithm, the proposed FL + LSTM + GA method significantly improves the pollution source positioning accuracy to 99.5% and reduces the average absolute error (MAE) of Gaussian model parameter estimation to 0.20.

## 1. Introduction

Air pollution poses a significant and pressing challenge for modern society. Its detrimental effects extend beyond the environment, encompassing glacier melting, extreme temperature variations, droughts, and floods. Moreover, air pollution has grave implications for human well-being and health. These consequences include cardiovascular disease and asthma [1], as well as pneumonia and influenza in the elderly [2]. Globally, approximately 1.1 billion people are exposed to unhealthy air each year, resulting in 7 million deaths [3,4]. To effectively combat air pollution, it is essential to accurately identify the sources of pollution. There are two main approaches for achieving precise pollutant localization: pollutant dispersion modeling through air dynamic simulation and extensive monitoring of air sensors.

The study of pollutant dispersion using physical modeling tracking methods has been conducted in recent years. Douglas et al. studied the dispersion of bioaerosols using an atmospheric dispersion model [5]. Parvez et al. proposed a local-scale dispersion model R-LINE to estimate the mass of road emission sources [6]. Monbureau et al. proposed a modification to the plume dispersion model that significantly improved the simulation of ground-level pollutant concentrations in buildings [7]. However, contaminant traceability approaches based on physical or statistical methods have some drawbacks. The approach proposed by Karl et al. [8] for estimating air pollutants using deterministic chemical transport models that incorporate meteorological, physical, and chemical processes is susceptible to uncertainties in emission quantities and chemical reactions. Meanwhile, methods for predicting pollutants by statistical models, such as multiple linear regression models [9] and geographically weighted regression models [10], often result from simplifying the complex relationships between air pollutant concentrations and predictor variables. Also, due to the lack of bandwidth [11], network security vulnerability issues [12], and insufficient data protection [13,14] in the traditional IoT context. We present a novel algorithm that integrates a federated learning (FL) framework and a Long Short-Term Memory (LSTM) model to enhance the Gaussian model through a retrospective approach. FL is widely used as one of the most promising distributed machine learning frameworks, enabling resource-constrained edge devices to collaboratively build shared global models [15]. In light of the increasing focus on data protection in both domestic and international legal frameworks, Federated Learning (FL) is emerging as a prominent machine learning technique to address the issue of data silos prevalent in modern times. Diverging from traditional federated learning approaches, our methodology utilizes genetic algorithms instead of cloud neural networks to accelerate the learning process and achieve iterative optimization.

With the rapid development of AI (Artificial Intelligence) and IoT (Internet of Things), it is widely considered that AIoT systems, such as intelligent light poles with detection modules, are also effectively used for air quality monitoring. However, these systems use monitoring data for air quality analysis offline and do not achieve real-time intelligent computing analysis with edge intelligence algorithms. The number of installations is constrained and unable to satisfy the intricate requirements of air pollution traceability due to the high cost of building, operating, and maintaining such fixed systems. Additionally, the currently common compact sensors are only functional for the surroundings around the instrument and are unable to adequately respond to the overall air quality in an area, particularly in cities where air pollution in cities is related to terrain and building height. An important problem arises when attempting to identify the source of contaminants at certain nodes. In terms of IoT technologies, Abbas et al. developed a smart meter based on Narrowband IoT [16]. Ejaz et al. investigated the problem of energy-efficient scheduling for smart homes and wireless power transmission for the Internet of Things in smart cities [17]. In addition to these applications, 5G and Wi-Fi can also be relevant to the operation of smart cities [18]. The theory of collaborative multi-intelligent body sensing has strong technical backing thanks to the convergence of edge computing and networked communication discovery. Edge computing has attracted increasing attention as an effective solution to address long latency issues and enhance existing network architectures [19]. The hypothesis of multi-intelligent body collaborative perception is well supported technically by the convergence of edge computing and networked communication discovery.

In this paper, we design an air pollutant traceability system based on EIPA networks. The contributions of this study are as follows:(1)The EIPA networks refer to a type of multi-agent network that combines Artificial Intelligence and the Internet of Things. In this network, intelligent light poles are installed with wireless sensors and edge computing capabilities, serving as stationary agent nodes.(2)In order to accurately track the location of air pollution, it is crucial to implement a sophisticated air quality monitoring system that can operate in various weather conditions. In this study, we propose the development of an Enhanced Intelligent Lamp-Post (EIPA) system, which involves integrating air quality sensors and edge computing technology into conventional lamp-posts. Additionally, we introduce a novel algorithm that enables the traceability of air pollutants through a collaborative learning approach involving cloud and edge computing.(3)Owing to the limited occurrence of air pollution events within industrial parks, the acquisition of an adequate amount of pollution data presents a challenge. To address this issue, we have developed a dataset using computational fluid dynamics (CFD) simulation. This dataset allows for the analysis of air pollutant data and the verification of the air pollutant traceability algorithm. The findings of our study demonstrate promising potential for the application of the EIPA system. Furthermore, we have made our data openly accessible to other researchers to facilitate further investigations in this field.(4)Air pollutants are a complex mixture of gases and particulate matter that can have detrimental effects on both the environment and human health. Major gaseous pollutants include nitrogen oxides (NO_x_), sulfur dioxide (SO_2_), carbon monoxide (CO), ozone (O_3_), and volatile organic compounds (VOCs), while particulate matter (PM) is classified by size as PM_10_ (diameter ≤ 10 μm) and PM_2.5_ (diameter ≤ 2.5 μm). These pollutants originate from diverse sources such as industrial emissions, vehicle exhaust, coal combustion, and natural processes [20].

PM_2.5_, in particular, has attracted significant attention due to its ability to penetrate deep into the respiratory system and enter the bloodstream, causing chronic respiratory and cardiovascular diseases [21]. The chemical composition of these pollutants varies widely depending on emission sources and atmospheric transformation processes, making accurate traceability a challenging task. In urban environments, the interaction between pollutants and meteorological conditions further complicates the dispersion patterns of pollutants.

Effective management and mitigation of air pollution require not only advanced monitoring technologies but also a comprehensive understanding of pollutant behavior in different environmental contexts. Recent studies have focused on developing integrated approaches combining real-time sensing, data-driven modeling, and physical simulations to improve the accuracy of pollution source identification and prediction [22]. Such approaches are crucial for formulating targeted pollution control strategies and protecting public health.

To address the challenges of air pollutant traceability in complex urban environments, this paper proposes an innovative approach that integrates federated learning, LSTM, and genetic algorithms within an Edge Intelligent Perception Agent (EIPA) network. The subsequent sections are organized as follows: Section 2 presents a comprehensive literature review, highlighting the limitations of existing physicochemical, statistical, and machine learning-based methods for pollution source localization. Section 3 details the proposed methodology, including the hardware architecture of the EIPA system, the federated learning framework, the LSTM model design, and the integration with genetic algorithms for parameter optimization. Section 4 describes the experimental setup, dataset generation using computational fluid dynamics (CFD) simulations, and the evaluation results comparing the proposed approach with traditional methods. Finally, Section 5 summarizes the key findings, discusses the practical implications of the research, and outlines potential directions for future work.

## 2. Literature Review

In this section, we will discuss previous research conducted on environmental detection sensor networks, which serves as the foundation for our proposed algorithm. Additionally, we will review relevant studies on the application of machine learning algorithms for the purpose of tracing and predicting air pollution. This review aims to provide readers with a comprehensive understanding of our proposed algorithm, which emphasizes the development of multi-agent intelligent perception networks and a dual-driven architecture that integrates knowledge data for environmental monitoring and air pollutant traceability.

### 2.1. Background

Since the last century, chemical industry parks have become significant production spaces for countries. However, they have also exerted immense pressure on the surrounding and regional environment. Ground-level concentrations of air pollutants in these areas are not always directly correlated with source emissions. This discrepancy poses a greater challenge for pollution accountability. For instance, pollutants released from chimneys at higher emission heights may be more diluted compared to those released at ground level [23]. Therefore, identifying the locations where pollutants disperse in complex industrial parks at ground level is a problem that needs to be addressed.

The turbulent flow of air masses in urban environments is often attributed to the complex spatial arrangement resulting from rapid urban development. Although high-precision air quality monitoring stations have been widely used, the concentration of pollutants generated in a particular place mainly depends on local emission sources and atmospheric flow conditions [24]. As a result, there may be some discrepancies between the precise air quality and the data obtained from the monitoring stations. One possible approach to addressing these challenges is through the implementation of a substantial quantity of affordable monitoring devices.

Given the urgency of real-world pollution monitoring, it is imperative that we have the capability to promptly address unforeseen instances of pollutant leakage. Additionally, there may be instances where certain enterprises engage in illicit practices of emitting pollutants. Consequently, it is essential for us to possess the ability to track pollutants in real-time in order to overcome these challenges.

### 2.2. Related Works

Effective prediction of air quality and tracking pollution sources are hot topics in interdisciplinary research. Different types of methods for predicting air quality and tracing air pollutants are worth reviewing and analyzing. The mainstream approaches include physicochemical models, statistical models, and machine learning algorithms [25]. However, there are still some difficulties to overcome, as indicated by the literature research. Physicochemical models incorporate the chemical properties of pollutants, including lateral and longitudinal diffusion coefficients, along with physical environment models such as atmospheric pressure, temperature, and humidity, to simulate the dispersion of pollutants. These models typically rely on Gaussian diffusion models. However, rapid urbanization has introduced complex geographic environments, resulting in inaccurate input parameters and posing challenges to the ability of traditional Gaussian models to provide accurate predictions. Zhu et al. [26] applied particle filters to the Gaussian diffusion model, continuously updating the diffusion coefficients by assimilating observed data into the model during the calculation process. They also proposed error propagation detection rules to improve the prediction accuracy of the Gaussian model. In the field of environmental analysis, statistical models are commonly used to evaluate the distribution of pollutants by analyzing their statistical characteristics, such as seasonal and spatial factors. However, the applicability of general distribution models is limited due to the diverse range of statistical properties associated with air pollutants, as well as the challenges in comprehensively understanding and incorporating these properties across various geographic environments. Williams et al. [27] borrowed methods from nonequilibrium statistical mechanics to construct a suitable super-statistical model for air pollution and improve the accuracy of risk estimation. In the current year, there has been significant adoption of machine learning and edge computing technologies, leading to the widespread utilization of LSTM or other hybrid models for predicting pollutant concentrations. Du et al. proposed a hybrid model based on CNN and BiLSTM to learn multivariate air quality data, using CNN to extract local trend features and spatially correlated features and BiLSTM to learn spatio-temporal dependence relationships. This approach accurately predicted PM_2.5_ concentrations. Muthukumar et al. [28] used graphical convolutional networks (GCN) and convolutional long and short-term memory (ConvLSTM) to learn the spatio-temporal dependencies of PM_2.5_ from remotely sensed satellite images and sensor data. To the best of our understanding, the majority of previous research has concentrated on tracing high-altitude pollutants over large spatial and temporal scales. However, the issue of pollutant leakage within small-scale urban campuses at low altitudes remains unresolved. Meanwhile, current pollutant traceability methods still rely on traditional dispersion models, which require detailed physical modeling of complete pollutant emission data [29].

An effective strategy for addressing the aforementioned challenges involves employing a dual-driven approach that combines physical modeling with big data learning. This approach holds considerable importance and offers numerous advantages for investigating the real-time dynamics of air pollutant traceability. The objective of this research is to devise a tracking algorithm that is dual-driven by big data learning and the Gaussian diffusion model. The specific focus is on tracing pollution sources in urban industrial parks.

## 3. Methodology

### 3.1. Hardware Architecture and Data Acquisition Methods

The EIPA network has been designed with the objective of providing a comprehensive artificial intelligence (AI) solution and facilitating the development of smart cities. The hardware architecture and data flow of the EIPA network are illustrated in Figure 1. The static nodes, represented by smart light poles, serve as the foundation for edge computing and smart sensing networks, enabling various applications in smart cities, including air pollution monitoring. The EIPA system can sense the urban environment autonomously and flexibly by collaborating with static nodes. The EIPA nodes will establish communication with the server in order to exchange information and receive commands for activating various functions. Additionally, a smart terminal application has been developed to operate on the EIPA network. This application serves as a control interface, enabling human operators to monitor and control various nodes. After obtaining the location of the contaminant dispersion center, the administrator can assess the leakage situation and take appropriate measures. In this paper, we mainly introduce the construction of the EIPA static node and its application algorithm.

Each static EIPA node is capable of autonomously selecting multi-modal sensors according to specific environmental monitoring requirements, thereby constructing a sensor array to fulfill diverse monitoring needs. The selection of sensors presented in Table 1 highlights some of the ones utilized in our study.

The DHT11 module is a sensor used for measuring temperature and humidity. It consists of an 8-bit microcontroller unit (MCU) that is connected to a resistive humidity sensing element and a Negative Temperature Coefficient (NTC) temperature measuring element. The module retrieves data by reading the value from the MCU.

The MQ-2, MQ-7, and MQ-137 sensors are part of a series of sensors that monitor gas composition. These sensors utilize tin dioxide as a gas-sensitive material, which becomes highly reactive when heated to its operating temperature. At this temperature, tin dioxide adsorbs oxygen from the surrounding air, leading to the formation of negatively charged oxygen ions. This process results in a reduction in electron density within the semiconductor, thereby increasing its resistance. When the sensor comes into contact with smoke, the concentration of smoke causes a change in the potential barrier at the intergranular boundary, resulting in a modification of surface conductivity. By utilizing this phenomenon, it becomes possible to gather information regarding the presence of smoke. The relationship between smoke concentration and electrical conductivity is directly proportional, while the relationship between smoke concentration and output resistance is inversely proportional. Consequently, a higher concentration of smoke leads to a stronger output analog signal. The MQ-2 sensor, known for its high sensitivity to alkane smoke and resistance to interference, is employed in conjunction with an ADC circuit to convert the voltage signal into a digital format. This digital signal is then further processed to obtain an accurate measurement of smoke concentration. The MQ-7 sensor utilizes temperature cycle detection methods to detect the concentration of carbon monoxide. It employs low temperature (1.5 V voltage heating) to detect carbon monoxide by measuring the conductivity of the sensor with the air containing carbon monoxide gas. The conductivity of the sensor increases as the concentration of carbon monoxide gas in the air rises. On the other hand, a high temperature (5.0 V heating) is used to eliminate any stray gas that may have been adsorbed at a low temperature. The MQ-137 sensor is specifically designed to detect ammonia gas in the surrounding environment. The conductivity of the sensor increases in proportion to the concentration of ammonia gas in the air. This information can be collected by the ADC circuit to accurately determine the concentration of ammonia gas.

The BMP388 air pressure sensor adopts piezoelectric pressure sensor technology, offering higher resolution and lower power consumption. The BMP388 barometric pressure sensor adopts piezoelectric pressure sensor technology with higher resolution and lower power consumption. It converts the voltage signal into a digital signal and then accurately measures the barometric pressure value using the ADC circuit.

After obtaining the environmental parameters, a Gaussian dispersion model can be used to fit the dispersion state of the pollutant. The equation of the Gaussian dispersion model is as follows:(1)$$Q=2\pi *C_{i}*V*\sigma_{y}*\sigma_{z}*e^{-\frac{y^{2}}{2{\sigma_{y}}^{2}}}*\left[e^{-\frac{(z-H{)}^{2}}{2{\sigma_{y}}^{2}}}+e^{-\frac{{\left(z+H\right)}^{2}}{2{\sigma_{z}}^{2}}}\right]$$
where *Q* represents the intensity of the field source pollutant, V denotes the measured wind speed, Ci represents the concentration of the pollutant detected at the ith light pole, and *σ_y_* and *σ_z_* are the diffusion coefficients of pollutants in the horizontal and vertical directions. H is the effective source height, y is the distance relative to the pollutant centerline in the perpendicular direction, and z is the height of the monitoring point.

The functions of *σ_y_* and *σ_z_* are presented:(2)$$\left\{\begin{array}{c} \sigma_{y}=a*x^{b}\\ \sigma_{z}=c*x^{d} \end{array}\right.$$
where x is the relative distance of the light pole in the downwind direction of the pollution source. Table 2 shows the reference values of parameters *a*, *b*, *c*, and *d*.

The traditional Gaussian model, however, is no longer suitable for modeling the dispersion of air pollutants within cities due to the high level of development of modern cities and the shading of pollutants by tall structures. We divide the metropolitan areas using intelligent light poles and independently update the Gaussian dispersion models in each area to address the issue of pollutant dispersion in contemporary cities.

### 3.2. Federated Learning Algorithm Design

In the context of federated learning (FL), intelligent terminals leverage their local data to train a deep learning model that is necessary for the central server. Subsequently, these intelligent terminals transmit the model parameters, as opposed to the raw data, to the server to facilitate cooperative learning tasks. Due to its ability to protect data privacy, Lee et al. applied federated learning (FL) to medical image analysis, specifically in the context of thyroid ultrasound image analysis [30]. Nehal Muthukumar discusses the use of FL in natural language processing [31] by incorporating the Amazon Review dataset. Wu et al. applied this approach to IoT-based human activity recognition [32] as a way to enhance security and privacy within the IoT context. The anonymity of air pollutant traceability research is not an issue, though. FL has the ability to reduce data connection bandwidth and excels at activities requiring real-time dynamic air pollutant traceability.

We have enhanced the conventional FedAVG algorithm to facilitate the implementation of distributed joint learning. In the original FedAVG approach, during each iteration, a central server chooses a fixed number of nodes (k) to distribute copies of the initial model parameters (vt). The selected static nodes update the local model wk,t, 0 = vt with the received model parameters. They will be trained locally using the environmental data detected by the sensors on the nodes, with Ω local iterations performed by the optimizer, after which each static node uploads the local model wk,t +1 = wk, t, Ω to the central server, which aggregates them to get a new global model [33] for the next training session.(3)$$v_{t+1}=\frac{1}{K}\Sigma w_{k,t+1}$$

In this case, the central server or other nodes never directly see the data on any other nodes, which effectively protects data privacy.

Our enhanced algorithm for tracking the origins of pollution from air pollution data incorporates a novel FL model that combines LSTM and GA. The complete workflow of the algorithm is illustrated in Figure 2. Each stationary node collects and preprocesses time series data using a sensor array integrated with a lamp pole. The edge computing module, equipped with the lamp pole, utilizes long-term and short-term memory modules to learn from extensive data. Afterward, the training model parameters and predicted values are uploaded to the central server, where the FedAVG, combined with the GA optimization program, is executed. The detailed design of the GA and the LSTM learning process is elaborated in Section 3.4.

The aim of the entire Federated Learning framework is to optimize the gain factor *γ_i_* for the Gaussian diffusion model and to make full use of the edge computing module on the smart light pole while reducing the burden on the central server. In an infinitely large unobstructed wind field, the concept of *γ_i_* is presented:(4)$$\gamma_{i}=\frac{C_{i}}{\left(\frac{\Sigma C}{{\left(\rho -R_{i}\right)}^{2}}\right)}$$
where Ci is the concentration of the pollutant detected by the ith static node, C is the concentration of the source, and ‘ρ-Ri’ is the distance from the source to the monitoring point of the ith static node.

However, in different urban environments, *γ_i_* varies with the topography, so for the intra-city pollution dispersion problem, we need to obtain a refined gain factor for small areas.

In the event of a release of contaminants, the stationary node promptly notifies the server of any significant changes in the concentration of the contaminant. Subsequently, the server distributes a copy of the parameters of the global model to all stationary nodes that have detected a change in contaminant concentration. To effectively accomplish our objective in a practical setting, it is necessary to incorporate additional processing from the server to the nodes. The server has the capability to communicate with both stationary nodes via the Internet or wireless networks. We make the assumption that the following scenarios will occur:− Increase: When there is a pollutant leak, the EIPA system will add all EIPA Static nodes Ci that detect a change in pollutant concentration to the server. The server will then issue an initial model and give each node an initial weight *t_ij_* for the current leaked pollutant, where *t_ij_* represents the weight of the jth pollutant detected by EIPA node Ci.− Decrease: Whenever an EIPA node detects a change in pollutant concentration less than *ε_j_*, the EIPA system will automatically remove the Static node from the server, along with the weight *t_ij_* of the node.− Periodic Response: At regular intervals, the EIPA system will broadcast an interrogation message to all nodes to ensure that all nodes are online and can be activated quickly for contaminant processing.

Upon receiving the global model parameters from the central server, the static node proceeds to update its local model and utilize the local data for training purposes. The input data consists of meteorological information along with pollutant concentrations. The air quality data is collected by sensor arrays placed at static EIPA nodes, which monitor several important air pollutants, including particulate matter (PM_10_ or PM_2.5_), ammonia (NH3), and carbon monoxide (CO). These pollutants are commonly employed in the calculation of the Air Quality Index (AQI) for specific locations. The static EIPA node simultaneously maintains a record of diverse meteorological parameters, such as relative humidity, atmospheric pressure, temperature, velocity, and wind direction. These parameters are utilized to predict the dispersion centers of pollutants. Subsequently, the collected data undergoes a data cleaning process to rectify any issues, such as ensuring data consistency and handling invalid or missing values.

### 3.3. LSTM Algorithm Design

Data-driven time series prediction methods have shown their effectiveness in a variety of industrial production. Imad Alawe et al. investigated the role of LSTM models in flow load prediction [34], while Shu et al. utilized LSTM models for interpersonal relationship recognition to address the challenge of recognizing human interactions in videos [35].

The LSTM model controls the discarding or addition of information through the gate to achieve the functions of forgetting and remembering. Figure 3 depicts a standard LSTM cell that has three gates: the forget gate, the input gate, and the output gate.

Forget gate: The forget gate is a function of the output ht-1 of the previous unit and the input *x_t_* of this unit as input, and its output is ft, which is used to control the degree to which the state of the previous unit is forgotten. The concept of ft is presented as follows:(5)$$f_{t}=\sigma \left(W_{f}*\left[h_{t-1},\;x_{t}\right]+\;b_{f}\right)$$
where *W_f_* is the weight matrix of the forget gate, and bf is the bias term of the forget gate. *W_f_* is a random initial value from 0 to 1, and the initial value of b is 0.

Input gate: The input gate and a tanh function work together to control the input of new information. The concept is presented as follows.(6)$$i_{t}=\sigma \left(W_{i}*\left[h_{t-1},\;x_{t}\right]+b_{i}\right)$$
where *W_i_* is the weight matrix of the forget gate, bi is the bias term of the oblivion gate. Wi is a random initial value from 0 to 1, and the initial value of bi is 0.

Output gate: The output gate *o_t_* is used to control the amount of the current cell state that is filtered out. The input state is activated first, and the output gate controls the amount of the input state that is filtered out. The concept of *o_t_* is presented as follows:(7)$$o_{t}=\sigma \left(W_{o}*\left[h_{t-1},\;x_{t}\right]+\;b_{o}\right)$$
where *W_o_* is the weight matrix of the forget gate, *b_o_* is the bias term of the oblivion gate. Wo is a random initial value from 0 to 1, and the initial value of *b_o_* is 0.

Although LSTM models have been used for large-scale air quality prediction, only a few LSTM models have been utilized for source dispersion traceability in chemical parks. In contrast to conventional air quality prediction methods, chemical parks face additional challenges, such as building occlusion and the need for timely traceability. To address these obstacles, we propose a novel approach that combines a Long Short-Term Memory (LSTM) model with a genetic algorithm.

We train the dataset by deploying a Long Short-Term Memory (LSTM) model on a distributed static EIPA node. In the previous section, we introduced our optimization object *γ_i_*. After conducting a preliminary analysis, we found that *γ_i_* is influenced by the wind field. However, its precise value is also affected by factors such as temperature, humidity, atmospheric pressure, and the value from the previous moment. Since the recurrent neural network (RNN) model overwrites its memory in an uncontrolled manner at each time step, with both gradient explosion and gradient disappearance, the LSTM model, on the other hand, modifies its memory in a more precise way: by using a special mechanism to selectively remember and update information, it is able to keep track of the information for a longer duration. Therefore, we introduce the LSTM model to train on *γ_i_*.

During the phase of training the model, the central server employs a random selection process to choose several static nodes. These nodes are then determined to be suitable through the addition and deletion methods discussed in the preceding section. Subsequently, the global model is distributed to the selected nodes. Upon receiving the model, each static node utilizes the environmental data collected from the smart light pole to train the model. Once the local training is completed, the node uploads the model weights to the central server. The server then aggregates these model weights to generate a new global model.

### 3.4. Genetic Algorithm Optimization

The genetic algorithm is a probabilistic technique used for global optimization in search. It emulates the processes of replication, crossover, and mutation that take place in natural selection and heredity. By initiating with an initial population, the algorithm progresses towards a more favorable region in the search space through random selection, crossover, and mutation operations. This results in the generation of individuals who are better suited to the environment. Through multiple iterations, the algorithm eventually converges to a group of individuals that are best adapted to the environment, thereby identifying the optimal solution.

A typical genetic algorithm (GA) is structured as follows: initially, each individual within the solution space is encoded using binary code. Subsequently, an adaptive degree function F is proposed. New individuals are generated within the search space through the utilization of selection, crossover, and variation operators. Simultaneously, in order to prevent convergence to a local optimum, the fitness is compromised. This means that individuals with higher fitness have a greater likelihood of being selected, while those with lower fitness have a reduced chance of being chosen. The optimal solution is achieved through multiple iterations. The process of a standard genetic algorithm is depicted in Algorithm 1.
**Algorithm 1: Genetic Algorithm****Input**: pop (initial populations), p_c_ (crossover probability), p_c_ (mutation probability)}, f (adaptability value), M (population size), g (number of iterations)**output**: Individual (x,y) with the lowest adaptationInitializing the population new_pop**Do**:  **Do**:     Randomly generate two random numbers between (0,1) and select two individuals     Randomly generate a random number l between (0,1)     if : l < p_c_         Crossover of two individual chromosomes to produce a new individual         Put two individuals into the new population new_pop     if : l < p_b_         Two individuals mutate to produce a new individual         Put two individuals into the new population new_pop  **until** |new_pop| = M  pop ← new_pop**until** The number of iterations reached or individual adaptation to reach f**return** (x,y)

In the model inference stage, the smart light pole will use the sensor detection data on the pole to reason about the cloud model. And upload the Gaussian correction parameters of the area where the smart light pole is located to the cloud, which will use the gain factor to locate the pollution source by genetic algorithm.

With the LSTM model, we are able to obtain the next moment of *γ_i_*, and combined with the next moment of monitoring values, we can obtain the fitted diffusion center of the pollution source through the genetic algorithm. The objective function F is presented as follows:(8)$$F_{min}=\Sigma {\left(\frac{C}{{\left({X-X}_{i}+x\right)}^{2}+{\left({Y-Y}_{i}+y\right)}^{2}}-C_{i}\right)}^{2}$$
where C is the concentration of the source, X and Y are the coordinates of the pollution source, Xi and Yi are the coordinates of the ith detection point, x and y are the coordinate compensation calculated by *γ_i_*, and Ci is the concentration value of the pollutant detected at the ith detection point.

In an effort to enhance the stability and resilience of Federated Learning during the assessment of the global model provided by static EIPA nodes, we present an optimization approach that leverages the real-time monitoring values of adjacent light poles for validation purposes. Following the completion of each training iteration, the light poles make predictions on the monitoring data of the three closest neighboring light poles and transmit these predictions to the central server. The central server then requests the actual monitoring data from the respective light poles and evaluates the training outcomes using this methodology. A concrete example illustrating our algorithm is depicted in Figure 4.

In this example, after completing one round of training, Node 1 will upload the detection data of the three smart light poles of Node 2, Node 3, and Node 4, which are predicted for the next moment. The central server will then request the current actual detection values from the above three light poles, compare them with the predicted data, update the Gaussian model, and retrain it.

## 4. Discussion

In the experimental section, the configuration and assessment criteria for the test dataset are initially introduced, followed by the design of several experiments aimed at validating the algorithm’s viability.

### 4.1. Dataset and Test Situation

Due to the rarity of large-scale pollutant leaks in the real world. We conducted simulations and collected data using OpenFOAM to verify the reliability of the algorithm. It presents a simulation of steady flow over buildings with varying inlet velocities, utilizing RANS-type turbulence modeling. The mesh is constructed using the snappyHexMesh tool, and the total number of elements is up to 190,000. The case simulates the atmospheric wind flow around a group of buildings, which could be part of an urban park. The mesh model of the urban wind field is shown in Figure 5. In this project, the simulation space measures 350 m in length, 280 m in width, and 140 m in height and contains various buildings, predominantly located in the central area.

The fluid solver used in this work is based on the simpleFoam solver of OpenFOAM, in which the following conservation equations of mass and momentum are solved:(9)$$\frac{\partial \rho }{\partial t}+\nabla \cdot \left(\rho v\right)=0$$(10)$$\frac{\partial \rho }{\partial t}+\nabla \cdot \left(\rho v\right)=0$$
where t is time, ρ is mixture density, v is velocity, p is pressure, S is the momentum source, and $\overset{̿}{\tau }$ is the stress tensor following Stokes’ hypothesis:(11)$$\overset{̿}{\tau }=-\frac{2}{3}\mu \left(\nabla \cdot v\right)\overset{̿}{I}+\mu \left[\nabla v+{\left(\nabla v\right)}^{T}\right]$$
where μ is the dynamic viscosity, $\overset{̿}{I}$ is the identity tensor. Most time, the stress tensor can be simplified as a turbulence model, such as the RAS or the LES model.

Tests are performed on a desktop computer equipped with an 8C16T 3.60GHz Intel (R) Core (TM) i9-9900K CPU and an RTX2080Ti. The programming language used is Python 3.9.1, based on TensorFlow. The software platform for tests is vs. Code. Due to external influencing conditions, such as different ambient temperatures on the chassis, the experimental results may fluctuate. In other words, the prediction of the pollutant dispersion center may vary within a small range for the same sample. The author selects the higher value as the final result, which is displayed in the table and figure.

Once the data has been trained, the model is subjected to testing, and the error rate is assessed using the Mean Absolute Error (MAE) evaluation. This evaluation involves measuring the error rate on both the test and validation sets. MAE is a technique used to evaluate the accuracy of a model by calculating the error. It quantifies the relative discrepancy between the predicted values and the actual detected values of the model. MAE is calculated by the following formula:(12)$$L_{MAE}=\frac{1}{N}\sum_{i=1}^{N}{\left(y_{i}-\hat{y_{i}}\right)}^{2}$$
where N is the number of samples, yi is the predicted Gaussian model correction coefficient, and yi is the Gaussian model correction coefficient of the ith sample. The relative position error is used to evaluate the prediction results by positioning accuracy. The objective function ACC is introduced(13)$$ACC=1-\frac{{\left(x-\bar{x}\right)}^{2}+{\left(y-\bar{y}\right)}^{2}}{{x_{t}}^{2}+{y_{t}}^{2}}$$
where x is the predicted x-axis coordinate; y is the predicted y-axis coordinate; $\bar{x}$ and $\bar{y}$ are the actual coordinates; and xt and yt are the building group length and width values.

### 4.2. Evaluation of OpenFOAM Generated Data

After completing the OpenFOAM setup, we conducted experiments on pollutant dispersion. We obtained a significant amount of data by deploying multiple detection points in the experimental area. Table 3 displays a selection of our pre-defined experiments and the corresponding pollutant data obtained through these experiments. Meanwhile, we used the collected data to build a dataset on pollutant dispersion in the influence of buildings.

In this example, we first performed a 400 s simulation of wind field dispersion in the experimental area. After achieving a uniform distribution of the wind field, we arranged the source dispersion center at (0, 140). To improve accuracy, we deployed 16 detection points in this complex and simulated the actual detection data within 2 min after the dispersion occurred. Table 4 displays the specific coordinate points and the actual detection data after 2 min simulated by OpenFOAM. Coordinates and values of detection points.

### 4.3. Evaluation of FL and LSTM Model

In this section, we design some experiments to evaluate the performance of our proposed algorithm. Federated Learning and LSTM models are used to estimate the parameters of the Gaussian model, which is crucial for calculating the center of pollutant diffusion. The data for training, validation, and testing are generated through the following process. Gaussian model parameters in different environments are collected through multiple simulation experiments in discrete-time scenarios to create a dataset. These parameters are then randomly and uniformly distributed to multiple users for training, with each user using 80% of the obtained data as the training set and the rest as the test set. Finally, the global model is updated by performing a weighted average aggregation and sending it to the central server after completing the training process. We conducted comparison experiments using the results of the traditional genetic algorithm, LSTM + genetic algorithm, and Federated Learning + LSTM + genetic algorithm. The results are shown in Table 5, where MAE represents the estimation accuracy of the parameters of the modified Gaussian model, and ACC represents the localization accuracy. Compared to the LSTM + genetic algorithm, the method incorporating the Federated Learning framework reduces the MAE value by 1.01 and improves accuracy by 2.8%. This result demonstrates that the Federated Learning framework can effectively improve performance.

In another experiment, we tested the same dataset using LSTM + genetic algorithm and Federated Learning + LSTM + genetic algorithm, respectively. Figure 6a shows the fitting effect of the dataset under LSTM, and Figure 6b shows the fitting effect of the dataset under FL + LSTM. It can be observed that the model fits the dataset more accurately when the Federated Learning framework is incorporated.

### 4.4. Evaluation of Method

In this section, we use a specific example to evaluate our algorithm in OpenFOAM. First, we completed the wind field modeling for 400 s. Then, we placed a spherical pollutant source with a diameter of 10 m at the coordinates (0, 140). We performed the pollutant dispersion simulation for 2 min. For data acquisition, we used the 16 points mentioned in Table 2. We obtained the pollutant concentration values at each detection point for a 2 min duration. The variation in each detection point, and the final predicted pollutant diffusion center by our algorithm is (1.92, 144.68) as shown in Figure 7a. We also compared the fit of the LSTM + genetic algorithm with that of the traditional genetic algorithm, as shown in Figure 7b,c. By comparison, we can see that this algorithm offers higher prediction accuracy, making it more convenient to detect with greater precision.

## 5. Conclusions

In this paper, we present a pollution source localization algorithm designed for EIPA systems in the context of smart cities. The algorithm utilizes the Federated Learning framework and the LSTM algorithm to track the dispersion centers of pollution sources. The static EIPA nodes serve as the computing platform, while the input data is obtained from the sensor networks of these static nodes. We discuss the application of this algorithm in tracking the diffusion center of air pollutants and compare it with the traditional genetic algorithm and the LSTM + genetic algorithm. The experimental results demonstrate that the proposed FL + LSTM + GA method achieves a localization accuracy of 99.5% with a mean absolute error (MAE) of only 0.20 for Gaussian model parameter estimation. This represents a significant improvement over both the LSTM + GA approach (96.7% accuracy, MAE = 1.21) and the traditional genetic algorithm (73.8% accuracy). These results confirm the effectiveness of integrating federated learning with LSTM and genetic algorithms for air pollutant traceability. The combination of the Federated Learning framework and the LSTM algorithm effectively locates the diffusion center of pollutants. Additionally, the EIPA system offers advantages such as dynamic monitoring, data visualization, high localization accuracy, and scalability. In future work, our goal is to enhance the speed and accuracy of localization by incorporating advanced AI algorithms and integrating additional edge computing units to cater to various scenarios.

## Figures and Tables

**Figure 1 sensors-25-06119-f001:**
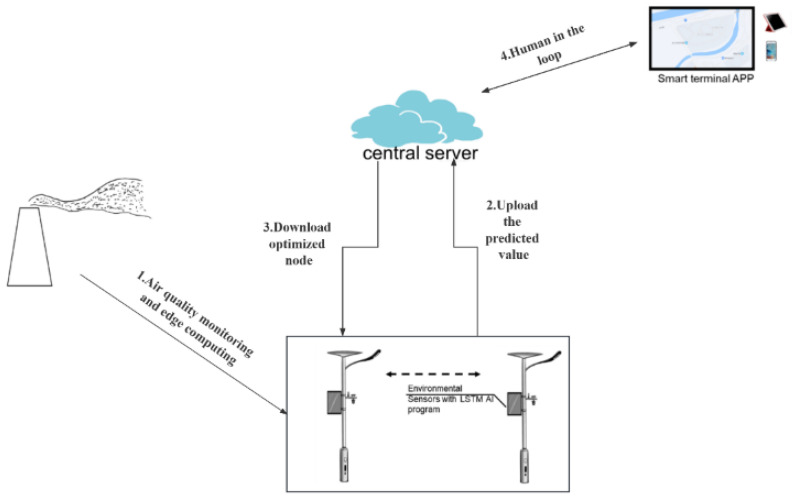
The overall structure of the EIPA system.

**Figure 2 sensors-25-06119-f002:**
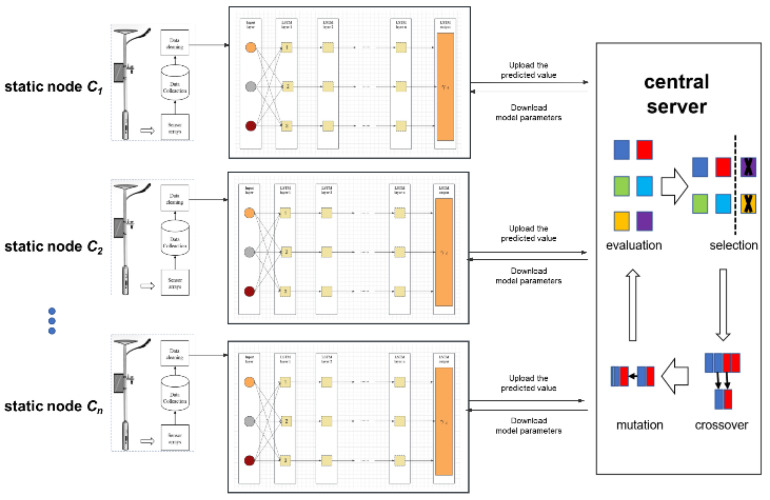
The flowchart of the proposed algorithm.

**Figure 3 sensors-25-06119-f003:**
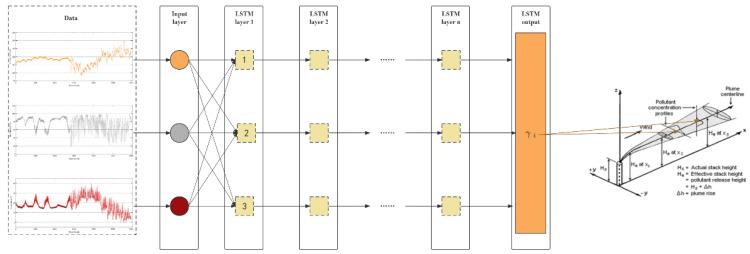
The LSTM model.

**Figure 4 sensors-25-06119-f004:**
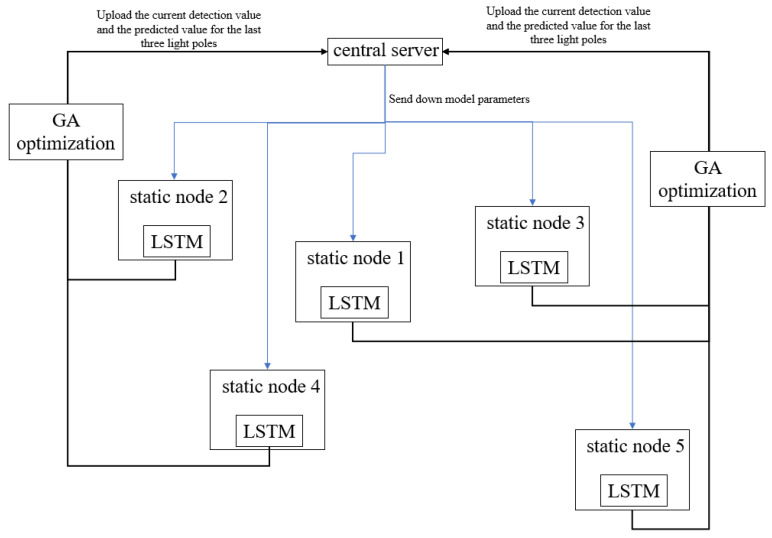
The optimization methods.

**Figure 5 sensors-25-06119-f005:**
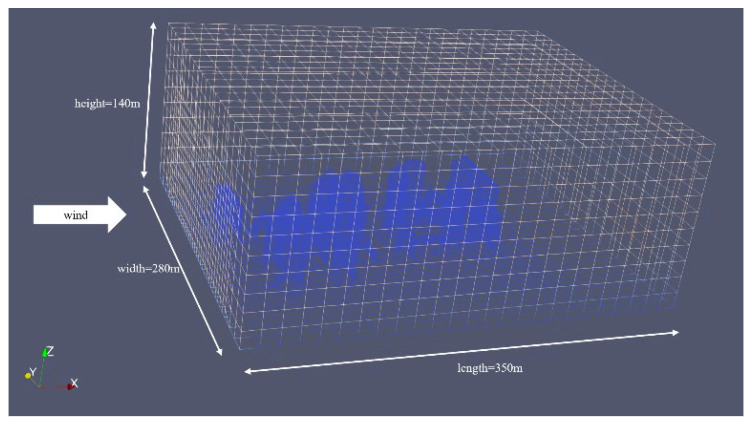
The mesh model of the urban wind field.

**Figure 6 sensors-25-06119-f006:**
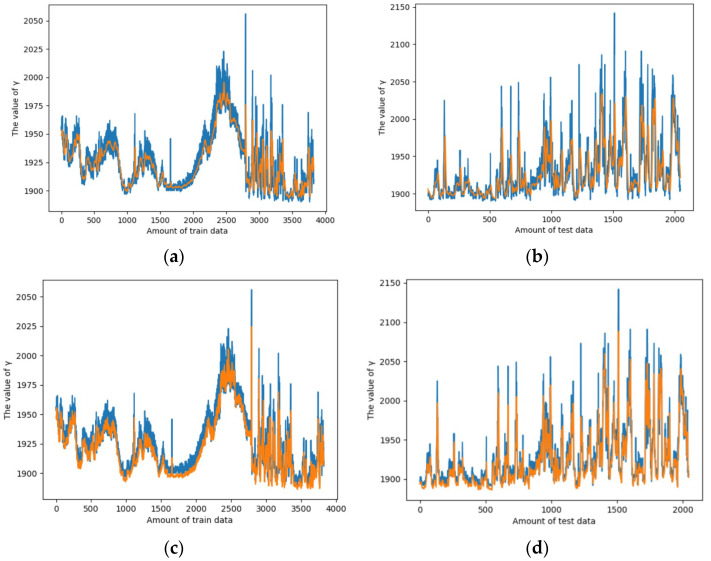
Training effect of the dataset under LSTM and FL + LSTM. (**a**) The fitting effect of the train data uder LSTM. (**b**) The fitting effect of the test data under LSTM. (**c**) The fitting effect of the train data under FL + LSTM. (**d**) The fitting effect of the test data under FL + LSTM.

**Figure 7 sensors-25-06119-f007:**
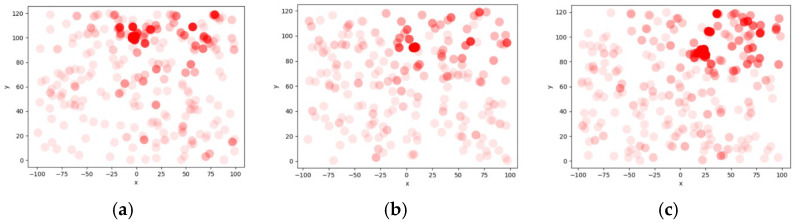
The results of ablation experiment comparison. (**a**) Prediction results by FL + LSTM + GA. (**b**) Prediction results by LSTM + GA. (**c**) Prediction results by GA.

**Table 1 sensors-25-06119-t001:** Sensors for static EIPA nodes.

NO.	Sensor Type	Sensor Function
1	DHT11	Temperature and humidity sensor
2	MQ-2	Smoke Sensor
3	MQ-7	CO sensor
4	MQ-137	NH3 sensor
5	BMP388	Air pressure sensor

**Table 2 sensors-25-06119-t002:** Atmospheric stability.

Level	a	b	c	d
A	0.52	0.865	0.28	0.90
B	0.371	0.866	0.23	0.85
C	0.209	0.897	0.22	0.80
D	0.123	0.905	0.20	0.76
E	0.098	0.902	0.15	0.73
F	0.065	0.902	0.12	0.67

**Table 3 sensors-25-06119-t003:** Detection point coordinates and detection values.

Wind Direction and Pollution Source Design	Pollutant Dispersion Experiment	Data We Obtained
Wind direction: from left to right The source of pollution: (0, 120)	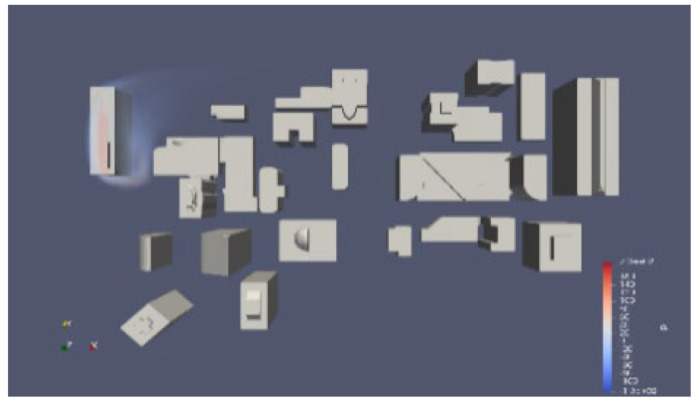	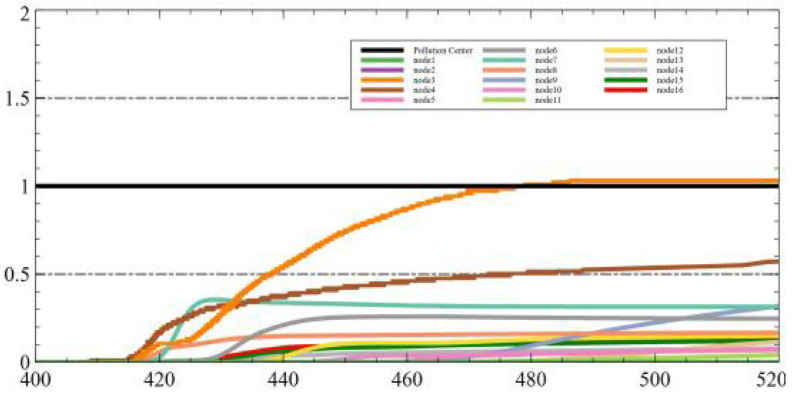
Wind direction: from left to right The source of pollution: (0, 80)	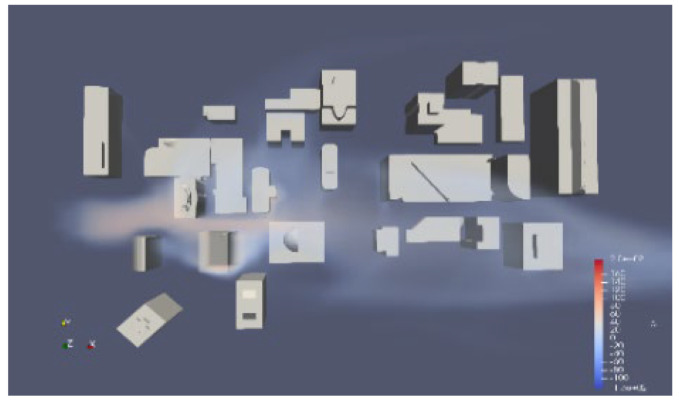	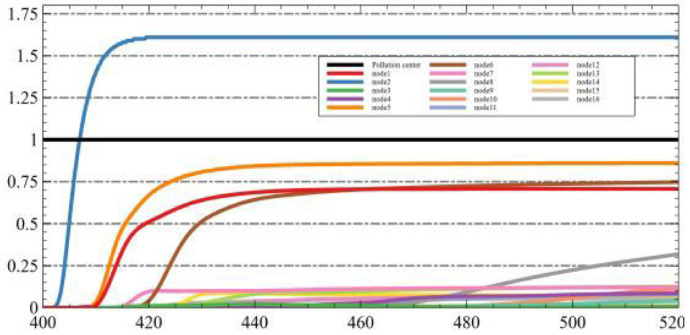
Wind direction: from left to right The source of pollution: (0, 40)	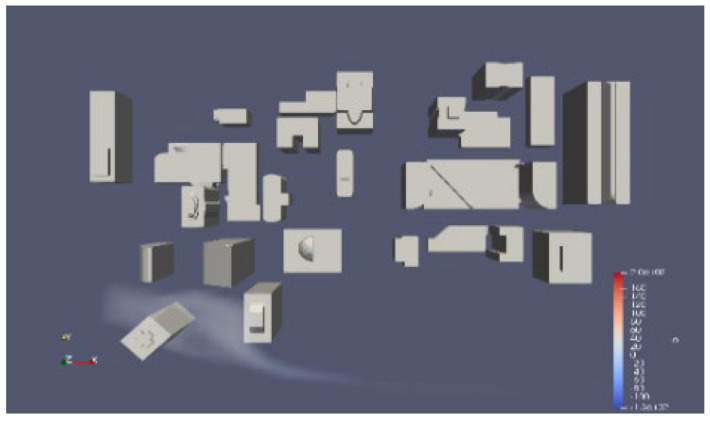	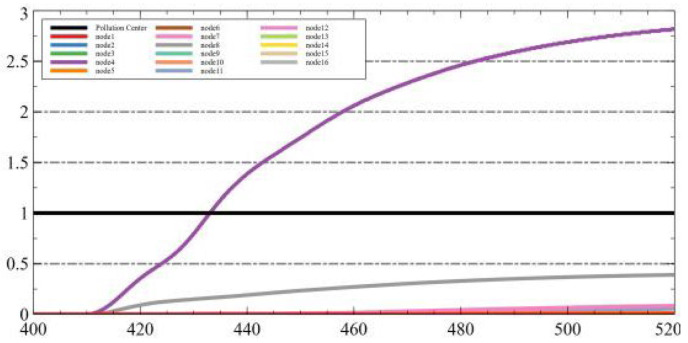

**Table 4 sensors-25-06119-t004:** Detection coordinates.

NO.	Coordinates	Pollutant Concentration Value
1	(40, 20)	0.014
2	(40, 60)	0.014
3	(40, 100)	1.199
4	(40, 140)	1.807
5	(100, 20)	0.110
6	(85, 60)	0.161
7	(100, 100)	0.846
8	(85, 140)	1.637
9	(135, 20)	0.102
10	(135, 60)	0.020
11	(135, 100)	0.375
12	(135, 140)	0.362
13	(200, 20)	0.014
14	(200, 60)	0.029
15	(215, 100)	0.013
16	(215, 140)	0.027

**Table 5 sensors-25-06119-t005:** Comparison experiment of our method.

Method	MAE	ACC
GA	—	73.8%
LSTM + GA	1.21	96.7%
FL + LSTM + GA	0.20	99.5%

## Data Availability

Data are contained within the article.

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
