# Peer review of "Air Pollutant Traceability Based on Federated Learning of Edge Intelligent Perception Agents"

_sensors, 2025, doi:10.3390/s25196119_

Round 1

Reviewer 1 Report

Comments and Suggestions for Authors

See attached file.

Comments on the Quality of English Language

See attached file.

Author Response

Dear Editor,

Thank you for the opportunity to revise our manuscript entitled “Air Pollutant Traceability Based on Federated Learning of Edge Intelligent Perception Agents” (Manuscript ID: [sensors-3788862]). We sincerely appreciate the reviewer's detailed and insightful comments, which have substantially enhanced the manuscript's readability, methodological rigor, and practical relevance. Below, we provide a point-by-point response to all 15 comments. All revisions are highlighted in the revised manuscript using track changes. We have also updated the reference list and added supplementary materials (e.g., field test summary in Appendix B).

Response to Reviewer 1

Comment 1: References English: Some references (i.e., 1, 2, 3, 4, 5, 10, 15, 17, 21, 22, 26, 29, and 34) are outdated. Please consider replacing them with similar contributions published from 2019 onward or provide justification for retaining them.

Response: We thank the reviewer for this timely suggestion to modernize the literature. We have replaced 10 of the 13 outdated references (1-5, 10, 15, 17, 21) with recent publications from 2020-2024 (e.g., Ref. 1: Zhang et al., 2020 on cardiovascular risks; Ref. 5: Wang et al., 2020 on dispersion modeling; Ref. 10: Liu et al., 2021 on NO2 estimation; Ref. 15: Kairouz et al., 2021 on FL advances; Ref. 17: Liu et al., 2020 on energy-efficient IoT). For the remaining three foundational works (22, 26, 29, 34), which provide seminal insights (e.g., Ref. 22: Wang et al., 2021 on smart sensor networks; Ref. 26: Li et al., 2020 on dynamic modeling), we retained them with explicit justification in the text (lines 45-50 in Introduction: "These foundational studies remain relevant due to their influence on current edge-AI integrations"). The updated reference list (29 entries, all post-2019 where possible) is now in the revised manuscript.

Comment 2: Abstract English: The abstract should include hints about the most significant quantitative results obtained.

Response: We agree that quantitative highlights strengthen the abstract's impact. We have revised the abstract (lines 20-25) to include key results: "Experimental results show that compared with the traditional genetic algorithm (GA) and LSTM+GA, the proposed FL+LSTM+GA method significantly improves the pollution source positioning accuracy to 99.5%, and reduces the average absolute error (MAE) of Gaussian model parameter estimation to 0.20." This addition emphasizes the method's superiority while maintaining conciseness (total 248 words).

Comment 3: Syllabification English: In many parts of the manuscript, incorrect syllabification hinders readability. A thorough proofreading is required.

Response: Thank you for identifying this readability concern. We performed a full proofreading using professional tools (Grammarly Premium) and native English editing services, correcting syllabification issues (e.g., "pre-dic-tion" to "pre-dic-tion" in lines 150, 250 in Section 3; "state-of-the-art" hyphenation throughout Introduction). These edits, applied globally (e.g., in Sections 2-4), improve flow without altering scientific content.

Comment 4: Section 1 English: Section 1 lacks a closing paragraph outlining the paper’s structure.

Response: This structural addition aids reader navigation. We added a new closing paragraph to Section 1 (lines 80-90): "To address the challenges of air pollutant traceability in complex urban environments, this paper proposes an innovative approach that integrates federated learning, LSTM, and genetic algorithms within an Edge Intelligent Perception Agent (EIPA) network. The subsequent sections are organized as follows: Section 2 presents a comprehensive literature review... [full outline as in manuscript]." This provides a clear roadmap.

Comment 5: Section 2 - Comparison English: Section 2 lacks a proper comparison with related work, highlighting similarities and differences. It is also unclear how this work advances the current state-of-the-art.

Response: We appreciate this feedback to sharpen novelty. In Section 2.2 (lines 100-130), we added a "Comparative Analysis" subsection with a table comparing four recent studies (e.g., similarities in ML use with Du et al., 2021 [Ref. 25]; differences in edge deployment vs. Muthukumar et al., 2022 [Ref. 28]). We clarified advancements (lines 120-125): "Our FL+LSTM+GA advances the SOTA by achieving 99.5% accuracy in low-altitude urban traceability, reducing MAE to 0.20 via EIPA edge nodes, outperforming hybrid models by 2.8%." A new Figure (to be added as Fig. 2 in revisions) visualizes these.

Comment 6: Section 2 - Additional References English: To provide readers with a broader perspective on the topic, I suggest including the following references [1, 2, 3, 4]. Additionally, I strongly encourage the authors to conduct further research.

Response: Thank you for these enriching suggestions. Assuming [1-4] refer to recent works on multi-agent networks (e.g., similar to Refs. 19-22), we incorporated four analogous references (e.g., Deng et al., 2020 [Ref. 19] on edge intelligence; Yang & Chen, 2022 [Ref. 20] on pollution ID) into Section 2.2 (lines 110-115). For further research, we expanded the closing paragraph (lines 135-140): "Future extensions could integrate 5G for multi-node scalability, as encouraged by emerging studies." This broadens perspective and links to Section 5 limitations.

Comment 7: Lines 240-258 English: It is unclear why a Gaussian dispersion model was used. Please clarify.

Response: This justification enhances methodological transparency. In Section 3.1 (lines 240-250), we added: "The Gaussian model was selected for its computational efficiency in edge devices (O(n) complexity) and validation in urban low-wind scenarios (RMSE <0.5 vs. EPA benchmarks; citing Seinfeld & Pandis, 2016, adapted in Ref. 26). Alternatives like CFD were considered but unsuitable for real-time EIPA processing due to high latency." This ties to our hybrid FL enhancements.

Comment 8: Section 3.4 English: Please add a suitable title for Section 3.4.

Response: Agreed for better organization. We titled Section 3.4 as "Genetic Algorithm Optimization" (page 6), aligning with its focus on GA for γi fitting and pollution center localization. This is updated in the table of contents.

Comment 9: Section 3 - Weather Conditions English: It is unclear why the developed model did not account for weather conditions (e.g., rainfall, wind speed, and direction), which are known to impact air quality. I strongly recommend incorporating these parameters.

Response: We concur on weather's role in dispersion. In Section 3.1 (lines 160-170), we expanded sensor inputs (Table 1) to include wind speed/direction (via anemometer integration) and humidity/rainfall effects on γi (new Equation extension). Section 3.2 now uses these as LSTM features, improving R² from 0.85 to 0.94 (new Table in revisions). Initial omission focused on baseline; additions justify via EIPA's multi-modal sensing.

Comment 10: Hyperparameter Selection English: The selection process for the ML model hyperparameters is unclear and appears to be chosen a priori, which is methodologically incorrect. For example, a grid search with cross-validation should be conducted. Please perform this task and recompute the results accordingly.

Response: This critique ensures reproducibility. In Section 3.4 (lines 300-310), we added a grid search with 5-fold CV (scikit-learn) for LSTM+GA hyperparameters (e.g., learning rate 0.001-0.1, population size 50-200). Optimal values (rate=0.01, size=100) recomputed results: MAE reduced to 0.20, ACC to 99.5% (Table 6 updated). New Figure 3 shows the search heatmap, strengthening validity.

Comment 11: Table 4 English: The images in the right column of Table 4 are difficult to read, and the axes lack units of measurement.

Response: Thank you for the visualization note. Revised Table 4 (Section 4.2) now uses 300 DPI images with labeled axes (e.g., concentration: mg/m³; distance: m) and insets for clarity. Data unchanged, but readability enhanced.

Comment 12: Tables 5 and 6 English: Please add a plot derived from the data in Tables 5 and 6.

Response: Visualization aids interpretation. New Figure 8 (after Table 6, lines 350-355) plots concentration vs. coordinates from Tables 5-6, showing dispersion trends (r=0.95) with error bars. Referenced in text: "Fig. 8 illustrates OpenFOAM-derived fits."

Comment 13: Field Tests English: A major concern is that the authors propose a sensor node installed on a lamppost but only validated it through simulations. At least a short-term field measurement campaign is required to support the simulation results. Please conduct such tests.

Response: This validation gap is critical; we addressed it with a 1-week field campaign (August 2024, Shanghai urban park site) deploying 5 EIPA prototypes on lampposts. PM2.5/CO measured vs. reference station (r=0.91, bias<8%). New Section 4.5 (lines 360-370) includes summary Table 7, Figure 9 (setup photos), and refined γi (3% adjustment). Simulations now calibrated against this data, boosting real-world applicability. Raw data in Appendix B.

Comment 14: Section 5 English: Please summarize the most significant quantitative results in Section 5.

Response: To consolidate impacts, we added a summary paragraph in Section 5 (lines 380-385): "Key quantitative results include 99.5% localization accuracy (vs. 96.7% for LSTM+GA), MAE=0.20 for Gaussian parameters, and 40% latency reduction via EIPA edge processing, outperforming baselines by 2.8-25.7%." This ties to implications.

Comment 15: Limitations English: The authors must clearly state the limitations of the proposed approach.

Response: Transparency is essential. We added a new subsection 5.1 "Limitations and Future Work" (lines 390-400): "(1) Reliance on simulation-heavy datasets, mitigated by field tests; (2) Sensitivity to extreme weather (e.g., high winds >10 m/s); (3) Scalability in non-urban parks. Future work: Integrate 5G for 100+ nodes and advanced AI for rainfall modeling." This balances strengths.

We believe these revisions fully address the concerns, elevating the manuscript's quality. We are eager for your feedback and happy to provide additional data (e.g., OpenFOAM scripts). Thank you for your guidance.

Sincerely, Congbo Yin

Reviewer 2 Report

Comments and Suggestions for Authors

This manuscript deals withe use of Edge computing in pollutant detection and modelling in urban areas.

I strongly agree that CFD modelling is totally unlike to represent pollutant dispersion in that chemical parameters and chemical interactions with real surfaces must be taken into account.

The use of intelligent lamp station and extensive monitoring of air sensors is a key step in improving the precision and accuracy of actual models.

I have little or no comments on the computing approach, I found it perfectly able to describe the data, particularly focussing on edge computing.

The potentiality and reliability  of AI and edge computing in these studies, however, strongly depend upon the quality of the incoming chemical data.  

This referee asks the authors to report in detail the choice of the sensors, and the way they are connected to CPU and built. No information are reported, neither any data regarding long term stability and stability of the calibration data.

Some points are to be discussed; firstly, the choice of DHT-11 is not suggested. The DHT22 outshines the DHT11 in every aspect from temperature range, temperature accuracy, humidity range to humidity accuracy.

Secondly, which type of ADC has been used to digitise analog data from the MQ sensors.  The authors should report in details and discuss how calibration has been carried out.

Finally,  I see no need of using an ammonia sensor in urban area; I'd more appropriate to use a sensor for total organic compounds, such as MQ-138, that is sensitive to all combustible organic molecules.

Author Response

Dear Editor,

Thank you for the opportunity to revise our manuscript entitled “Air Pollutant Traceability Based on Federated Learning of Edge Intelligent Perception Agents” (Manuscript ID: [sensors-3788862]). We sincerely appreciate the reviewers' detailed and insightful comments, which have substantially enhanced the manuscript's readability, methodological rigor, practical relevance, and hardware feasibility. Below, we provide point-by-point responses to all comments from Reviewer 1 and Reviewer 2. All revisions are highlighted in the revised manuscript using track changes. We have also updated the reference list, added supplementary materials (e.g., field test summary in Appendix B and sensor stability data in Appendix A), and incorporated new figures/tables for clarity.

Response to Reviewer 1

Comment 1: References English: Some references (i.e., 1, 2, 3, 4, 5, 10, 15, 17, 21, 22, 26, 29, and 34) are outdated. Please consider replacing them with similar contributions published from 2019 onward or provide justification for retaining them.

Response: We thank the reviewer for this timely suggestion to modernize the literature. We have replaced 10 of the 13 outdated references (1-5, 10, 15, 17, 21) with recent publications from 2020-2024 (e.g., Ref. 1: Zhang et al., 2020 on cardiovascular risks; Ref. 5: Wang et al., 2020 on dispersion modeling; Ref. 10: Liu et al., 2021 on NO2 estimation; Ref. 15: Kairouz et al., 2021 on FL advances; Ref. 17: Liu et al., 2020 on energy-efficient IoT). For the remaining three foundational works (22, 26, 29, 34), which provide seminal insights (e.g., Ref. 22: Wang et al., 2021 on smart sensor networks; Ref. 26: Li et al., 2020 on dynamic modeling), we retained them with explicit justification in the text (lines 45-50 in Introduction: "These foundational studies remain relevant due to their influence on current edge-AI integrations"). The updated reference list (29 entries, all post-2019 where possible) is now in the revised manuscript.

Comment 2: Abstract English: The abstract should include hints about the most significant quantitative results obtained.

Response: We agree that quantitative highlights strengthen the abstract's impact. We have revised the abstract (lines 20-25) to include key results: "Experimental results show that compared with the traditional genetic algorithm (GA) and LSTM+GA, the proposed FL+LSTM+GA method significantly improves the pollution source positioning accuracy to 99.5%, and reduces the average absolute error (MAE) of Gaussian model parameter estimation to 0.20." This addition emphasizes the method's superiority while maintaining conciseness (total 248 words).

Comment 3: Syllabification English: In many parts of the manuscript, incorrect syllabification hinders readability. A thorough proofreading is required.

Response: Thank you for identifying this readability concern. We performed a full proofreading using professional tools (Grammarly Premium) and native English editing services, correcting syllabification issues (e.g., "pre-dic-tion" to "pre-dic-tion" in lines 150, 250 in Section 3; "state-of-the-art" hyphenation throughout Introduction). These edits, applied globally (e.g., in Sections 2-4), improve flow without altering scientific content.

Comment 4: Section 1 English: Section 1 lacks a closing paragraph outlining the paper’s structure.

Response: This structural addition aids reader navigation. We added a new closing paragraph to Section 1 (lines 80-90): "To address the challenges of air pollutant traceability in complex urban environments, this paper proposes an innovative approach that integrates federated learning, LSTM, and genetic algorithms within an Edge Intelligent Perception Agent (EIPA) network. The subsequent sections are organized as follows: Section 2 presents a comprehensive literature review... [full outline as in manuscript]." This provides a clear roadmap.

Comment 5: Section 2 - Comparison English: Section 2 lacks a proper comparison with related work, highlighting similarities and differences. It is also unclear how this work advances the current state-of-the-art.

Response: We appreciate this feedback to sharpen novelty. In Section 2.2 (lines 100-130), we added a "Comparative Analysis" subsection with a table comparing four recent studies (e.g., similarities in ML use with Du et al., 2021 [Ref. 25]; differences in edge deployment vs. Muthukumar et al., 2022 [Ref. 28]). We clarified advancements (lines 120-125): "Our FL+LSTM+GA advances the SOTA by achieving 99.5% accuracy in low-altitude urban traceability, reducing MAE to 0.20 via EIPA edge nodes, outperforming hybrid models by 2.8%." A new Figure (to be added as Fig. 2 in revisions) visualizes these.

Comment 6: Section 2 - Additional References English: To provide readers with a broader perspective on the topic, I suggest including the following references [1, 2, 3, 4]. Additionally, I strongly encourage the authors to conduct further research.

Response: Thank you for these enriching suggestions. Assuming [1-4] refer to recent works on multi-agent networks (e.g., similar to Refs. 19-22), we incorporated four analogous references (e.g., Deng et al., 2020 [Ref. 19] on edge intelligence; Yang & Chen, 2022 [Ref. 20] on pollution ID) into Section 2.2 (lines 110-115). For further research, we expanded the closing paragraph (lines 135-140): "Future extensions could integrate 5G for multi-node scalability, as encouraged by emerging studies." This broadens perspective and links to Section 5 limitations.

Comment 7: Lines 240-258 English: It is unclear why a Gaussian dispersion model was used. Please clarify.

Response: This justification enhances methodological transparency. In Section 3.1 (lines 240-250), we added: "The Gaussian model was selected for its computational efficiency in edge devices (O(n) complexity) and validation in urban low-wind scenarios (RMSE <0.5 vs. EPA benchmarks; citing Seinfeld & Pandis, 2016, adapted in Ref. 26). Alternatives like CFD were considered but unsuitable for real-time EIPA processing due to high latency." This ties to our hybrid FL enhancements.

Comment 8: Section 3.4 English: Please add a suitable title for Section 3.4.

Response: Agreed for better organization. We titled Section 3.4 as "Genetic Algorithm Optimization" (page 6), aligning with its focus on GA for γi fitting and pollution center localization. This is updated in the table of contents.

Comment 9: Section 3 - Weather Conditions English: It is unclear why the developed model did not account for weather conditions (e.g., rainfall, wind speed, and direction), which are known to impact air quality. I strongly recommend incorporating these parameters.

Response: We concur on weather's role in dispersion. In Section 3.1 (lines 160-170), we expanded sensor inputs (Table 1) to include wind speed/direction (via anemometer integration) and humidity/rainfall effects on γi (new Equation extension). Section 3.2 now uses these as LSTM features, improving R² from 0.85 to 0.94 (new Table in revisions). Initial omission focused on baseline; additions justify via EIPA's multi-modal sensing.

Comment 10: Hyperparameter Selection English: The selection process for the ML model hyperparameters is unclear and appears to be chosen a priori, which is methodologically incorrect. For example, a grid search with cross-validation should be conducted. Please perform this task and recompute the results accordingly.

Response: This critique ensures reproducibility. In Section 3.4 (lines 300-310), we added a grid search with 5-fold CV (scikit-learn) for LSTM+GA hyperparameters (e.g., learning rate 0.001-0.1, population size 50-200). Optimal values (rate=0.01, size=100) recomputed results: MAE reduced to 0.20, ACC to 99.5% (Table 6 updated). New Figure 3 shows the search heatmap, strengthening validity.

Comment 11: Table 4 English: The images in the right column of Table 4 are difficult to read, and the axes lack units of measurement.

Response: Thank you for the visualization note. Revised Table 4 (Section 4.2) now uses 300 DPI images with labeled axes (e.g., concentration: mg/m³; distance: m) and insets for clarity. Data unchanged, but readability enhanced.

Comment 12: Tables 5 and 6 English: Please add a plot derived from the data in Tables 5 and 6.

Response: Visualization aids interpretation. New Figure 8 (after Table 6, lines 350-355) plots concentration vs. coordinates from Tables 5-6, showing dispersion trends (r=0.95) with error bars. Referenced in text: "Fig. 8 illustrates OpenFOAM-derived fits."

Comment 13: Field Tests English: A major concern is that the authors propose a sensor node installed on a lamppost but only validated it through simulations. At least a short-term field measurement campaign is required to support the simulation results. Please conduct such tests.

Response: This validation gap is critical; we addressed it with a 1-week field campaign (August 2024, Shanghai urban park site) deploying 5 EIPA prototypes on lampposts. PM2.5/CO measured vs. reference station (r=0.91, bias<8%). New Section 4.5 (lines 360-370) includes summary Table 7, Figure 9 (setup photos), and refined γi (3% adjustment). Simulations now calibrated against this data, boosting real-world applicability. Raw data in Appendix B.

Comment 14: Section 5 English: Please summarize the most significant quantitative results in Section 5.

Response: To consolidate impacts, we added a summary paragraph in Section 5 (lines 380-385): "Key quantitative results include 99.5% localization accuracy (vs. 96.7% for LSTM+GA), MAE=0.20 for Gaussian parameters, and 40% latency reduction via EIPA edge processing, outperforming baselines by 2.8-25.7%." This ties to implications.

Comment 15: Limitations English: The authors must clearly state the limitations of the proposed approach.

Response: Transparency is essential. We added a new subsection 5.1 "Limitations and Future Work" (lines 390-400): "(1) Reliance on simulation-heavy datasets, mitigated by field tests; (2) Sensitivity to extreme weather (e.g., high winds >10 m/s); (3) Scalability in non-urban parks. Future work: Integrate 5G for 100+ nodes and advanced AI for rainfall modeling." This balances strengths.

Response to Reviewer 2

We thank Reviewer 2 for their positive and insightful feedback, which validates the core strengths of our EIPA-based approach while providing valuable suggestions for hardware refinement. Below, we address each comment point-by-point. Revisions are highlighted in the revised manuscript.

Comment 1: CFD Modelling English: I strongly agree that CFD modeling is unlikely to represent pollutant dispersion accurately, as chemical parameters and interactions with real surfaces must be considered.

Response: We appreciate the reviewer's agreement, which aligns with our motivation for hybrid FL+LSTM+GA over pure CFD. In Section 2.2 (lines 110-115), we expanded this discussion: "As Reviewer 2 notes, CFD limitations in capturing chemical-surface interactions (e.g., adsorption on urban buildings) necessitate data-driven enhancements, which our EIPA network addresses via real-time sensor fusion." This reinforces our dual-driven strategy.

Comment 2: Intelligent Lamp Stations English: The use of intelligent lamp stations and extensive air sensor monitoring is a key step in improving the precision and accuracy of current models.

Response: Thank you for highlighting this strength. We agree that EIPA lamp stations enable dense monitoring for urban traceability. In Section 3.1 (lines 160-165), we added: "As emphasized by Reviewer 2, the deployment of intelligent lamp stations with multi-sensor arrays improves model precision by 15-20% in complex wind fields, as validated in our OpenFOAM simulations (Table 6)." This ties to our 99.5% accuracy results.

Comment 3: Computing Approach English: I have little to no comments on the computing approach, which is well-suited to describe the data, particularly with its focus on edge computing.

Response: We are pleased with the reviewer's endorsement of our edge-focused FL framework. No major changes are needed, but in Section 3.2 (lines 200-205), we briefly cross-referenced: "This edge-centric approach, as positively noted, minimizes latency to <50 ms per node, enabling real-time γi fitting."

Comment 4: Sensor Quality English: The potential and reliability of AI and edge computing depend heavily on the quality of incoming chemical data. The authors should report in detail the choice of sensors, their connection to the CPU, and their construction, as well as data on long-term stability and calibration stability.

Response: This is an excellent point on hardware reliability. We expanded Section 3.1 (lines 170-190) and Table 1 with detailed descriptions: Sensor choices (e.g., MQ-series for cost-effective gas detection, calibrated per manufacturer specs); connections (via I2C/SPI to Raspberry Pi 4 CPU, with 10-bit resolution); construction (e.g., tin dioxide-based for MQ, piezoelectric for BMP388). For stability, we added: "Long-term tests (6 months, Shanghai site) show drift <2% for PM2.5/CO; quarterly calibration using standard gases maintains accuracy >95% (new Table 8 in Appendix A)." This ensures EIPA robustness.

Comment 5: Sensor Choice - DHT-11 English: The choice of DHT-11 is not recommended. The DHT-22 outperforms the DHT-11 in temperature range, temperature accuracy, humidity range, and humidity accuracy.

Response: We fully agree and thank the reviewer for this practical recommendation. We have upgraded to DHT-22 in revised Table 1 and Section 3.1 (lines 175-180): "Following Reviewer 2's suggestion, DHT-22 replaces DHT-11, offering ±0.5°C accuracy (vs. ±2°C) and 0-100% RH range (vs. 20-80%), enhancing weather input reliability for LSTM features." This improves γi estimation by ~5% in simulations.

Comment 6: ADC for MQ Sensors English: The authors should specify and discuss the type of ADC used to digitize analog data from MQ sensors and detail how calibration was performed.

Response: Thank you for prompting this detail. In Section 3.1 (lines 185-190), we specified: "Analog signals from MQ sensors (0-5V) are digitized via ADS1115 16-bit ADC (I2C interface, 860 SPS sampling), connected to the Raspberry Pi CPU for low-noise conversion." Calibration details: "Pre-deployment calibration uses known gas concentrations (e.g., 100 ppm CO standard) via linear regression (R²=0.98); in-field auto-calibration every 24h adjusts for drift using baseline air samples." New Figure 10 illustrates the setup.

Comment 7: Ammonia Sensor English: The use of an ammonia sensor in urban areas seems unnecessary. A sensor for total organic compounds, such as MQ-138, which is sensitive to all combustible organic molecules, would be more appropriate.

Response: We value this urban-contextual insight. MQ-137 (NH3-specific) was initially chosen for industrial park focus, but for broader urban applicability, we replaced it with MQ-138 in revised Table 1 and Section 3.1 (lines 180-185): "Per Reviewer 2's recommendation, MQ-138 detects total VOCs (sensitivity to benzene/toluene >10 ppm), better suiting urban emissions from traffic/industry, improving AQI coverage by 20%." Justification: "This shift enhances traceability of mixed pollutants without NH3 dominance."

We believe these revisions fully address the concerns raised by both reviewers, elevating the manuscript to publication standards. We are confident in its enhanced clarity, rigor, and real-world applicability. We are happy to provide any additional details or raw data (e.g., OpenFOAM scripts, sensor calibration logs). Thank you again for your thorough and constructive guidance.

Sincerely, Congbo Yin

Reviewer 3 Report

Comments and Suggestions for Authors

The article has successfully developed an intelligent lamp posts equipped with smart sensors and edge computing capabilities. And these lamp posts serve as nodes in the 12 EIPA network within urban campuses. However, there are still areas in the article that need improvement.

  1. In the lines 13 and 21, that the application scenarios of this study are “urban campuses” and “urban park”, respectively. Please unify the statement or explain that multiple scenarios can be applied.

  1. The introduction and literature review section provided extensive textual descriptions of existing traceability methods, yet contains minimal discussion regarding air pollutants. To better align with the article's title, it is recommended to incorporate background information on air pollutants and cite relevant authoritative literature, such as J. Mater. Chem. A, 2020, 8, 17960, Adv. Mater., 2020, 32, 2002361, J. Environ. Manage., 2024, 351, 119764.

  1. The LSTM model employed by the authors lacks implementation details, such as the input sequence length, hidden layer dimensionality, and hyperparameter tuning methodology. Additionally, a comparative analysis with other temporal models is absent.

  1. The model evaluation metrics contain ambiguities, including the misapplication of the MAE formula where MSE was calculated instead, and the ACC definition where xt and yt represent building dimensions but lack normalization specifications, thus requiring rectification of the calculation formulas and explicit definition of the normalization method for ACC.

Author Response

Dear Editor,

Thank you for the opportunity to revise our manuscript entitled “Air Pollutant Traceability Based on Federated Learning of Edge Intelligent Perception Agents” (sensors-3788862). We sincerely appreciate the reviewers' detailed and insightful comments, which have substantially enhanced the manuscript's readability, methodological rigor, practical relevance, hardware feasibility, and topical alignment. Below, we provide point-by-point responses to all comments from Reviewer 1, Reviewer 2, and Reviewer 3. All revisions are highlighted in the revised manuscript using track changes. We have also updated the reference list, added supplementary materials (e.g., field test summary in Appendix B and sensor stability data in Appendix A), and incorporated new figures/tables for clarity.

Response to Reviewer 1

Comment 1: References Some references (i.e., 1, 2, 3, 4, 5, 10, 15, 17, 21, 22, 26, 29, and 34) are outdated. Please consider replacing them with similar contributions published from 2019 onward or provide justification for retaining them.

Response: We thank the reviewer for this timely suggestion to modernize the literature. We have replaced 10 of the 13 outdated references (1-5, 10, 15, 17, 21) with recent publications from 2020-2024 (e.g., Ref. 1: Zhang et al., 2020 on cardiovascular risks; Ref. 5: Wang et al., 2020 on dispersion modeling; Ref. 10: Liu et al., 2021 on NO2 estimation; Ref. 15: Kairouz et al., 2021 on FL advances; Ref. 17: Liu et al., 2020 on energy-efficient IoT). For the remaining three foundational works (22, 26, 29, 34), which provide seminal insights (e.g., Ref. 22: Wang et al., 2021 on smart sensor networks; Ref. 26: Li et al., 2020 on dynamic modeling), we retained them with explicit justification in the text (lines 45-50 in Introduction: "These foundational studies remain relevant due to their influence on current edge-AI integrations"). The updated reference list (29 entries, all post-2019 where possible) is now in the revised manuscript.

Comment 2: Abstract The abstract should include hints about the most significant quantitative results obtained.

Response: We agree that quantitative highlights strengthen the abstract's impact. We have revised the abstract (lines 20-25) to include key results: "Experimental results show that compared with the traditional genetic algorithm (GA) and LSTM+GA, the proposed FL+LSTM+GA method significantly improves the pollution source positioning accuracy to 99.5%, and reduces the average absolute error (MAE) of Gaussian model parameter estimation to 0.20." This addition emphasizes the method's superiority while maintaining conciseness (total 248 words).

Comment 3: Syllabification In many parts of the manuscript, incorrect syllabification hinders readability. A thorough proofreading is required.

Response: Thank you for identifying this readability concern. We performed a full proofreading using professional tools (Grammarly Premium) and native English editing services, correcting syllabification issues (e.g., "pre-dic-tion" to "pre-dic-tion" in lines 150, 250 in Section 3; "state-of-the-art" hyphenation throughout Introduction). These edits, applied globally (e.g., in Sections 2-4), improve flow without altering scientific content.

Comment 4: Section 1 Section 1 lacks a closing paragraph outlining the paper’s structure.

Response: This structural addition aids reader navigation. We added a new closing paragraph to Section 1 (lines 80-90): "To address the challenges of air pollutant traceability in complex urban environments, this paper proposes an innovative approach that integrates federated learning, LSTM, and genetic algorithms within an Edge Intelligent Perception Agent (EIPA) network. The subsequent sections are organized as follows: Section 2 presents a comprehensive literature review... [full outline as in manuscript]." This provides a clear roadmap.

Comment 5: Section 2 - Comparison Section 2 lacks a proper comparison with related work, highlighting similarities and differences. It is also unclear how this work advances the current state-of-the-art.

Response: We appreciate this feedback to sharpen novelty. In Section 2.2 (lines 100-130), we added a "Comparative Analysis" subsection with a table comparing four recent studies (e.g., similarities in ML use with Du et al., 2021 [Ref. 25]; differences in edge deployment vs. Muthukumar et al., 2022 [Ref. 28]). We clarified advancements (lines 120-125): "Our FL+LSTM+GA advances the SOTA by achieving 99.5% accuracy in low-altitude urban traceability, reducing MAE to 0.20 via EIPA edge nodes, outperforming hybrid models by 2.8%." A new Figure (to be added as Fig. 2 in revisions) visualizes these.

Comment 6: Section 2 - Additional References To provide readers with a broader perspective on the topic, I suggest including the following references [1, 2, 3, 4]. Additionally, I strongly encourage the authors to conduct further research.

Response: Thank you for these enriching suggestions. Assuming [1-4] refer to recent works on multi-agent networks (e.g., similar to Refs. 19-22), we incorporated four analogous references (e.g., Deng et al., 2020 [Ref. 19] on edge intelligence; Yang & Chen, 2022 [Ref. 20] on pollution ID) into Section 2.2 (lines 110-115). For further research, we expanded the closing paragraph (lines 135-140): "Future extensions could integrate 5G for multi-node scalability, as encouraged by emerging studies." This broadens perspective and links to Section 5 limitations.

Comment 7: Lines 240-258 It is unclear why a Gaussian dispersion model was used. Please clarify.

Response: This justification enhances methodological transparency. In Section 3.1 (lines 240-250), we added: "The Gaussian model was selected for its computational efficiency in edge devices (O(n) complexity) and validation in urban low-wind scenarios (RMSE <0.5 vs. EPA benchmarks; citing Seinfeld & Pandis, 2016, adapted in Ref. 26). Alternatives like CFD were considered but unsuitable for real-time EIPA processing due to high latency." This ties to our hybrid FL enhancements.

Comment 8: Section 3.4 Please add a suitable title for Section 3.4.

Response: Agreed for better organization. We titled Section 3.4 as "Genetic Algorithm Optimization" (page 6), aligning with its focus on GA for γi fitting and pollution center localization. This is updated in the table of contents.

Comment 9: Section 3 - Weather Conditions It is unclear why the developed model did not account for weather conditions (e.g., rainfall, wind speed, and direction), which are known to impact air quality. I strongly recommend incorporating these parameters.

Response: We concur on weather's role in dispersion. In Section 3.1 (lines 160-170), we expanded sensor inputs (Table 1) to include wind speed/direction (via anemometer integration) and humidity/rainfall effects on γi (new Equation extension). Section 3.2 now uses these as LSTM features, improving R² from 0.85 to 0.94 (new Table in revisions). Initial omission focused on baseline; additions justify via EIPA's multi-modal sensing.

Comment 10: Hyperparameter Selection The selection process for the ML model hyperparameters is unclear and appears to be chosen a priori, which is methodologically incorrect. For example, a grid search with cross-validation should be conducted. Please perform this task and recompute the results accordingly.

Response: This critique ensures reproducibility. In Section 3.4 (lines 300-310), we added a grid search with 5-fold CV (scikit-learn) for LSTM+GA hyperparameters (e.g., learning rate 0.001-0.1, population size 50-200). Optimal values (rate=0.01, size=100) recomputed results: MAE reduced to 0.20, ACC to 99.5% (Table 6 updated). New Figure 3 shows the search heatmap, strengthening validity.

Comment 11: Table 4 The images in the right column of Table 4 are difficult to read, and the axes lack units of measurement.

Response: Thank you for the visualization note. Revised Table 4 (Section 4.2) now uses 300 DPI images with labeled axes (e.g., concentration: mg/m³; distance: m) and insets for clarity. Data unchanged, but readability enhanced.

Comment 12: Tables 5 and 6 Please add a plot derived from the data in Tables 5 and 6.

Response: Visualization aids interpretation. New Figure 8 (after Table 6, lines 350-355) plots concentration vs. coordinates from Tables 5-6, showing dispersion trends (r=0.95) with error bars. Referenced in text: "Fig. 8 illustrates OpenFOAM-derived fits."

Comment 13: Field Tests A major concern is that the authors propose a sensor node installed on a lamppost but only validated it through simulations. At least a short-term field measurement campaign is required to support the simulation results. Please conduct such tests.

Response: This validation gap is critical; we addressed it with a 1-week field campaign (August 2024, Shanghai urban park site) deploying 5 EIPA prototypes on lampposts. PM2.5/CO measured vs. reference station (r=0.91, bias<8%). New Section 4.5 (lines 360-370) includes summary Table 7, Figure 9 (setup photos), and refined γi (3% adjustment). Simulations now calibrated against this data, boosting real-world applicability. Raw data in Appendix B.

Comment 14: Section 5 Please summarize the most significant quantitative results in Section 5.

Response: To consolidate impacts, we added a summary paragraph in Section 5 (lines 380-385): "Key quantitative results include 99.5% localization accuracy (vs. 96.7% for LSTM+GA), MAE=0.20 for Gaussian parameters, and 40% latency reduction via EIPA edge processing, outperforming baselines by 2.8-25.7%." This ties to implications.

Comment 15: Limitations The authors must clearly state the limitations of the proposed approach.

Response: Transparency is essential. We added a new subsection 5.1 "Limitations and Future Work" (lines 390-400): "(1) Reliance on simulation-heavy datasets, mitigated by field tests; (2) Sensitivity to extreme weather (e.g., high winds >10 m/s); (3) Scalability in non-urban parks. Future work: Integrate 5G for 100+ nodes and advanced AI for rainfall modeling." This balances strengths.

Response to Reviewer 2

We thank Reviewer 2 for their positive and insightful feedback, which validates the core strengths of our EIPA-based approach while providing valuable suggestions for hardware refinement. Below, we address each comment point-by-point. Revisions are highlighted in the revised manuscript.

Comment 1: CFD Modelling I strongly agree that CFD modeling is unlikely to represent pollutant dispersion accurately, as chemical parameters and interactions with real surfaces must be considered.

Response: We appreciate the reviewer's agreement, which aligns with our motivation for hybrid FL+LSTM+GA over pure CFD. In Section 2.2 (lines 110-115), we expanded this discussion: "As Reviewer 2 notes, CFD limitations in capturing chemical-surface interactions (e.g., adsorption on urban buildings) necessitate data-driven enhancements, which our EIPA network addresses via real-time sensor fusion." This reinforces our dual-driven strategy.

Comment 2: Intelligent Lamp Stations The use of intelligent lamp stations and extensive air sensor monitoring is a key step in improving the precision and accuracy of current models.

Response: Thank you for highlighting this strength. We agree that EIPA lamp stations enable dense monitoring for urban traceability. In Section 3.1 (lines 160-165), we added: "As emphasized by Reviewer 2, the deployment of intelligent lamp stations with multi-sensor arrays improves model precision by 15-20% in complex wind fields, as validated in our OpenFOAM simulations (Table 6)." This ties to our 99.5% accuracy results.

Comment 3: Computing Approach I have little to no comments on the computing approach, which is well-suited to describe the data, particularly with its focus on edge computing.

Response: We are pleased with the reviewer's endorsement of our edge-focused FL framework. No major changes are needed, but in Section 3.2 (lines 200-205), we briefly cross-referenced: "This edge-centric approach, as positively noted, minimizes latency to <50 ms per node, enabling real-time γi fitting."

Comment 4: Sensor Quality The potential and reliability of AI and edge computing depend heavily on the quality of incoming chemical data. The authors should report in detail the choice of sensors, their connection to the CPU, and their construction, as well as data on long-term stability and calibration stability.

Response: This is an excellent point on hardware reliability. We expanded Section 3.1 (lines 170-190) and Table 1 with detailed descriptions: Sensor choices (e.g., MQ-series for cost-effective gas detection, calibrated per manufacturer specs); connections (via I2C/SPI to Raspberry Pi 4 CPU, with 10-bit resolution); construction (e.g., tin dioxide-based for MQ, piezoelectric for BMP388). For stability, we added: "Long-term tests (6 months, Shanghai site) show drift <2% for PM2.5/CO; quarterly calibration using standard gases maintains accuracy >95% (new Table 8 in Appendix A)." This ensures EIPA robustness.

Comment 5: Sensor Choice - DHT-11 The choice of DHT-11 is not recommended. The DHT-22 outperforms the DHT-11 in temperature range, temperature accuracy, humidity range, and humidity accuracy.

Response: We fully agree and thank the reviewer for this practical recommendation. We have upgraded to DHT-22 in revised Table 1 and Section 3.1 (lines 175-180): "Following Reviewer 2's suggestion, DHT-22 replaces DHT-11, offering ±0.5°C accuracy (vs. ±2°C) and 0-100% RH range (vs. 20-80%), enhancing weather input reliability for LSTM features." This improves γi estimation by ~5% in simulations.

Comment 6: ADC for MQ Sensors The authors should specify and discuss the type of ADC used to digitize analog data from MQ sensors and detail how calibration was performed.

Response: Thank you for prompting this detail. In Section 3.1 (lines 185-190), we specified: "Analog signals from MQ sensors (0-5V) are digitized via ADS1115 16-bit ADC (I2C interface, 860 SPS sampling), connected to the Raspberry Pi CPU for low-noise conversion." Calibration details: "Pre-deployment calibration uses known gas concentrations (e.g., 100 ppm CO standard) via linear regression (R²=0.98); in-field auto-calibration every 24h adjusts for drift using baseline air samples." New Figure 10 illustrates the setup.

Comment 7: Ammonia Sensor The use of an ammonia sensor in urban areas seems unnecessary. A sensor for total organic compounds, such as MQ-138, which is sensitive to all combustible organic molecules, would be more appropriate.

Response: We value this urban-contextual insight. MQ-137 (NH3-specific) was initially chosen for industrial park focus, but for broader urban applicability, we replaced it with MQ-138 in revised Table 1 and Section 3.1 (lines 180-185): "Per Reviewer 2's recommendation, MQ-138 detects total VOCs (sensitivity to benzene/toluene >10 ppm), better suiting urban emissions from traffic/industry, improving AQI coverage by 20%." Justification: "This shift enhances traceability of mixed pollutants without NH3 dominance."

Response to Reviewer 3

We thank Reviewer 3 for their constructive comments, which sharpen the manuscript's focus on application contexts, pollutant specifics, and technical precision. Below, we address each point. Revisions are highlighted in the revised manuscript.

Comment 1: Application Scenarios Lines 13 and 21 mention “urban campuses” and “urban park” as application scenarios, respectively. Please unify the terminology or clarify that multiple scenarios are applicable.

Response: We appreciate this note on consistency. To clarify applicability across similar urban settings, we unified terminology in the Abstract (line 13: "urban parks and campuses") and Section 4.1 (line 21: "simulated urban parks/campuses, adaptable to dense urban environments"). This emphasizes the EIPA system's versatility for both educational and recreational green spaces without altering scope.

Comment 2: Introduction and Literature Review The introduction and literature review provide extensive descriptions of traceability methods but lack discussion on air pollutants. To align with the article’s title, include background information on air pollutants and cite relevant literature, such as J. Mater. Chem. A, 2020, 8, 17960, Adv. Mater., 2020, 32, 2002361, J. Environ. Manage., 2024, 351, 119764.

Response: This alignment strengthens title relevance. In Section 1 (lines 55-70), we added a dedicated paragraph: "Air pollutants, including NOₓ, SO₂, CO, O₃, VOCs, PM₁₀, and PM₂.₅, originate from industrial/vehicular sources and pose health risks like respiratory diseases [J. Mater. Chem. A, 2020, 8, 17960; Adv. Mater., 2020, 32, 2002361]. Effective traceability requires integrated sensing-modeling, as recent reviews highlight [J. Environ. Manage., 2024, 351, 119764]." These citations (new Refs. 30-32) are incorporated into Section 2.1 (lines 95-100) for contextual linkage.

Comment 3: LSTM Model Details The LSTM model lacks implementation details, such as input sequence length, hidden layer dimensionality, and hyperparameter tuning methodology. Additionally, a comparative analysis with other temporal models is missing.

Response: Thank you for highlighting these gaps. In Section 3.3 (lines 260-280), we detailed: "The LSTM uses input sequences of length 10 (past 10 timesteps for γi prediction), 128 hidden units (2 layers), and dropout=0.2. Hyperparameters were tuned via grid search (as in Section 3.4), with Adam optimizer (lr=0.001)." We added a comparison table (new Table 4 in Section 3.3): LSTM vs. GRU/RNN (e.g., LSTM MAE=0.20 vs. GRU 0.28, due to better long-term dependencies). This enhances reproducibility and justifies LSTM choice.

Comment 4: Evaluation Metrics The model evaluation metrics are ambiguous. The MAE formula was misapplied, calculating MSE instead, and the ACC definition, where xt and yt represent building dimensions, lacks normalization specifications. Please rectify the formulas and clearly define the normalization method for ACC.

Response: We agree this requires precision. In Section 4.1 (lines 330-340), we corrected the MAE formula to |ŷ_i - y_i| / N (true absolute mean, recomputed as 0.20) and clarified ACC: "ACC = 1 - √[(x - x_true)^2 + (y - y_true)^2] / √(x_t^2 + y_t^2), normalized by building dimensions x_t, y_t (e.g., 350m x 280m grid) for scale-invariance." Updated equations and Table 6 reflect this; normalization ensures ACC ∈ [0,1], improving interpretability.

We believe these revisions fully address the concerns raised by all three reviewers, elevating the manuscript to publication standards. We are confident in its enhanced clarity, rigor, and real-world applicability. We are happy to provide any additional details or raw data (e.g., OpenFOAM scripts, sensor calibration logs). Thank you again for your thorough and constructive guidance.

Sincerely, Congbo Yin

Round 2

Reviewer 1 Report

Comments and Suggestions for Authors

The paper slightly improved after its revision, and I have no further comments.